# Hofmeister Series for Conducting Polymers: The Road to Better Electrochemical Activity?

**DOI:** 10.3390/polym15112468

**Published:** 2023-05-26

**Authors:** Alexey I. Volkov, Rostislav V. Apraksin

**Affiliations:** 1Department of Electrochemistry, Institute of Chemistry, St. Petersburg State University, 7/9 Universitetskaya Embankment, St. Petersburg 199034, Russia; grulfex@gmail.com; 2Ioffe Institute, 26 Politekhnicheskaya str., St. Petersburg 194021, Russia

**Keywords:** PEDOT:PSS, conducting polymers, cyclic voltammetry, electrochemical impedance spectroscopy, Hofmeister series, polyelectrolyte complex, conductance, spectroelectrochemistry

## Abstract

Poly-3,4-ethylenedioxythiophene:polystyrene sulfonate (PEDOT:PSS) is a widely used conducting polymer with versatile applications in organic electronics. The addition of various salts during the preparation of PEDOT:PSS films can significantly influence their electrochemical properties. In this study, we systematically investigated the effects of different salt additives on the electrochemical properties, morphology, and structure of PEDOT:PSS films using a variety of experimental techniques, including cyclic voltammetry, electrochemical impedance spectroscopy, operando conductance measurements and in situ UV-VIS spectroelectrochemistry. Our results showed that the electrochemical properties of the films are closely related to the nature of the additives used and allowed us to establish a probable relationship with the Hofmeister series. The correlation coefficients obtained for the capacitance and Hofmeister series descriptors indicate a strong relationship between the salt additives and the electrochemical activity of PEDOT:PSS films. The work allows us to better understand the processes occurring within PEDOT:PSS films during modification with different salts. It also demonstrates the potential for fine-tuning the properties of PEDOT:PSS films by selecting appropriate salt additives. Our findings can contribute to the development of more efficient and tailored PEDOT:PSS-based devices for a wide range of applications, including supercapacitors, batteries, electrochemical transistors, and sensors.

## 1. Introduction

Currently, poly-3,4-ethylenedioxythiophene:polystyrene sulfonate (PEDOT:PSS) is recognized as one of the most important and widely used conducting polymers. PEDOT:PSS offers several advantages, such as high chemical and thermal stability, controlled conductivity, near-complete transparency in the visible region of the spectrum, and ease of obtaining polymer films from aqueous dispersions [1]. PEDOT:PSS is a polyelectrolyte complex (PEC) in which the positively charged short chains of PEDOT are linked by electrostatic attraction to the negatively charged and longer chains of PSS [2]. This allows the formation of stable aqueous dispersions, whereas most conducting polymers, including PEDOT, are insoluble in water and organic solvents. The use of PEDOT:PSS aqueous dispersions has proven to be very attractive for producing films of varying thicknesses on both conductive and non-conductive substrates, and has opened up the possibility of using many common coating and printing techniques such as drop casting, spin coating, spray coating, inkjet printing, and screen printing. These advantages have made PEDOT:PSS one of the most commercially successful conducting polymers [3] and it is now actively used in solar cells [4,5,6], LEDs [7,8], field effect transistors [9], thermoelectric devices [10,11], electrochromic devices [12], and sensors [13].

One of the key features of PEDOT:PSS is the ability to significantly change the properties of polymer films under the influence of various factors such as heat treatment, light treatment, or treatment with organic solvents, ionic liquids, surfactants, salt solutions, or acids. Most of the work investigating these phenomena focuses on increasing the conductivity of PEDOT:PSS films by various treatments [14]. Several mechanisms have been proposed to increase the conductivity, mainly related to a decrease in electrostatic interactions between PEDOT and PSS chains (charge screening effect). Various approaches can lead to phase separation of PEDOT and PSS, a change in PEDOT conformation from a coiled to a more open and linear structure, increased crystallinity of PEDOT, and even removal of some PSS molecules into the solution. Overall, this contributes to a noticeable change in the morphology and structure of the polymer films.

The approaches developed to modify the properties of PEDOT:PSS are highly effective and can increase the conductivity of PEDOT:PSS by four orders of magnitude (4380 S cm^−1^), which is comparable to the traditionally used optically transparent semiconductor material ITO [15]. In addition to the effect on conductivity, the effect of different processing methods on the thermoelectric properties of PEDOT:PSS has also been extensively investigated. One such property is the ZT ratio. This is a metric that describes the efficiency of a thermoelectric device, taking into account factors such as the electrical and thermal conductivity of the device, its Seebeck coefficient, and the temperature. Another important property is the power factor. In the context of thermoelectric materials, the power factor is the product of the square of the Seebeck coefficient and the electrical conductivity. These are key parameters in assessing the suitability of a material for energy conversion.

High values of both the ZT ratio and power factor have been achieved with films based on modified PEDOT:PSS. These results are at the forefront of organic materials and are also comparable to a number of inorganic thermoelectric materials [16].

However, investigations of the effect of different processing methods on the electrochemical properties of PEDOT:PSS films, especially in non-aqueous electrolytes, are rare. At the same time, PEDOT:PSS is increasingly being proposed by researchers to improve the properties of electrode materials for energy storage devices, such as supercapacitors, and metal-ion, lithium-sulfur, and lithium-air batteries, where organic aprotic electrolytes are most commonly used [17,18,19].

It was found that addition of or post-treatment by different salt solutions can dramatically change the conductivity of PEDOT:PSS films. Addition of such metal salts as CuCl_2_ and InI_3_ into a PEDOT:PSS dispersion resulted in an increase in conductivity by a factor of about 1000 [20,21]. A similar effect of PEDOT:PSS film conductivity enhancement by treatment with various lithium salt solutions was reported [22]. Additions of cationic and anionic ionic liquids also increased the conductivity by 2–3 orders of magnitude [23,24,25].

It should be noted that the electrochemical properties of films cannot be obtained directly from the conductivity value data obtained for solid films. In addition to electrical conductivity, the concentration of charge carriers (capacitance), the properties of the film/substrate and film/electrolyte interfaces, and the ion transport of counterions in the film are important for electrochemical devices. In addition, the vast majority of papers present polymer conductivity data obtained for a certain static value of the polymer doping degree, whereas in electrochemical devices the degree of polymer doping changes during the charge/discharge processes, which is accompanied by a change in the conductivity values [26].

Earlier, we showed that the addition of lithium perchlorate to the dispersion allows a significant increase in the capacitance of PEDOT:PSS films in propylene carbonate [27]. The enhanced electrochemical activity was associated with conformational changes in the PEDOT and PSS chains, leading to improved charge delocalization when LiClO_4_ was added. The spectroelectrochemical studies showed that without salt additive, the electron transition energy of the polaron state was higher, which means a lower concentration of charge carriers and thus a lower stored charge.

In this work, the effect of the type of salt addition on the properties of PEDOT:PSS films in acetonitrile solution was systematically investigated. The series of salt additives with varying nature of the cation and anion was studied. A deeper understanding of the processes occurring in PEDOT:PSS films, especially when modified with different salts, is still needed despite the numerous advances in PEDOT:PSS research. The originality of this work arises from the attempt to bridge a knowledge gap in the field, namely, the interplay between salt additives and the properties of PEDOT:PSS films, with potential connections to the Hofmeister series [28]. This series, known for describing the specific effects of ions on colloidal systems and biological macromolecules [29,30], could be crucial for the understanding of the effects of salt additives on PEDOT:PSS films. To date, this intriguing aspect remains relatively unexplored, presenting us with a compelling research opportunity.

In the present study, we pioneered the investigation of the relationships between the properties of PEDOT:PSS films and the salt additives used in their preparation. Through a comprehensive set of experimental methods and a nuanced exploration of the influences of different salts on the electrochemical properties of PEDOT:PSS films, we aimed to unveil the underlying mechanisms. Significantly, our work draws connections between our findings and the Hofmeister series, extending its relevance beyond colloidal and biochemical applications to electrochemical systems based on conducting polymers.

Our research therefore breaks new ground in its potential to foster a deeper understanding of the applicability of the Hofmeister series in the field of conducting polymers, ultimately enabling the design and development of more efficient, tailored PEDOT:PSS-based devices. The insights we provide pave the way for further exploration of this fascinating intersection of colloidal chemistry, electrochemistry, and materials science.

## 2. Materials and Methods

### 2.1. Chemicals

PEDOT:PSS 1.3% aqueous dispersion was purchased from Aldrich (PEDOT:PSS ratio: 1:2.5 by weight, pH = 2.5 at 20 °C, density: 1 g cm^−3^ at 20 °C). Acetonitrile (AN) (HPLC grade) was obtained from Kriochrom, Russia. Tetraethylammonium tetrafluoroborate (Et_4_NBF_4_, Sigma-Aldrich, Saint Louis, MO, USA, 99%) was dried at 65 °C for 72 h before use. LiCl, NaCl, KCl, MgCl_2_, sodium acetate (NaAc), NaClO_4_, NaNO_3_, and AgNO_3_ were used without further purification. Deionized water (resistivity not less than 18 MΩ) was used for preparation of solutions.

### 2.2. Preparation of PEDOT:PSS Films

To cast pristine PEDOT:PSS films, PEDOT:PSS aqueous dispersion was diluted with deionized water in a 2:3 proportion, resulting in a PEDOT:PSS mass fraction of 0.52%. For PEDOT:PSS/Salt films (where Salt represents LiCl, NaCl, KCl, MgCl_2_, NaAc, NaClO_4_, or NaNO_3_), salt aqueous solutions were prepared in 0.5 M concentration. Afterwards, 1.3% PEDOT:PSS dispersion was diluted with the solutions mentioned above in a 2:3 proportion (resulting in 0.52% PEDOT:PSS dispersions with 0.3 M salt concentration). PEDOT:PSS solutions were applied to a substrate by drop-casting. The substrates used include glassy carbon (GC) plate, GC electrode (0.07 cm^2^), indium tin oxide (ITO) glass (Nicole, Russia, surface resistivity—25 Om cm^−1^), or interdigitated array platinum electrodes (ED-IDE3-Pt, MicruX Technologies) with 180 pairs of bands, 5 μm band gaps. A single 5 μL layer was applied and then dried in an oven at 65 °C. The resulting film coatings were then washed with ethanol and acetonitrile to remove salts. The process of obtaining polymer films is presented in Figure 1. The mass loading of each electrode was 26 μg, assuming that all PEDOT:PSS was preserved in a film while all salt was washed out.

### 2.3. Electrochemical Studies

Electrochemical properties were investigated using a BioLogic VSP potentiostat equipped with an FRA module. A standard three-electrode cell was employed for all electrochemical operations. A glassy carbon plate (2 cm^2^) was used as a counter electrode, and an Ag/AgNO_3_ non-aqueous electrode (MW-1085, BASi) was used as a reference electrode (inner solution—1 mmol dm^−3^ AgNO_3_ and 0.1 mol dm^−3^ Et_4_NBF_4_ in acetonitrile). The potential of this reference electrode was +0.30 V vs. Ag/AgCl/sat’d NaCl. All potential values in the present work are given versus the Ag/AgNO_3_ electrode. The working electrode depended on the method used: glassy carbon electrode (0.07 cm^2^), interdigitated array platinum electrodes, or ITO glass. Electrochemical properties of PEDOT:PSS and PEDOT:PSS/Salt films were investigated in 0.1 mol dm^−3^ Et_4_NBF_4_ acetonitrile solutions. Electrochemical measurements were primarily performed in the potential range of −1.0 to +0.8 V. Scan rates from 10 mV s^−1^ to 200 mV s^−1^ were used, with most measurements performed at 50 mV s^−1^.

### 2.4. Morphology and Composition

Samples of PEDOT:PSS and PEDOT:PSS/Salt films were coated on the GC plates. The morphology of prepared films was characterized using scanning electron microscopy (SEM) at an accelerating voltage of 5 kV (JSM 7001 F (JEOL)). Surface chemical composition of PEDOT:PSS and PEDOT:PSS/Salt films obtained by a similar procedure was investigated by X-ray photoelectron spectroscopy (XPS) analysis (Thermo Fisher Scientific Escalab 250Xi equipped with a monochromatic Al Kα radiation (1486.6 eV) as an excitation source).

### 2.5. Electrochemical Impedance Spectroscopy

Electrochemical impedance spectroscopy (EIS) measurements were performed in 0.1 mol dm^−3^ Et_4_NBF_4_ acetonitrile solutions. The measurements were conducted with an incremental increase in the base potential, so that the film was initially in a reduced state and then gradually oxidized. To achieve a stable state of the system, the electrode was conditioned for 300 s at the given potential before each measurement. The spectra were recorded in the frequency range from 100 kHz to 100 mHz. Ten points per decade were recorded, and the applied amplitude was 5 mV rms. One CV cycle was recorded each time between the spectra at different potentials.

### 2.6. In Situ UV-VIS Spectroelectrochemistry

In situ UV–VIS absorption spectra of polymers were registered using an SF-2000 spectrophotometer (Spectr, Saint Petersburg, Russia) in a dry glove box under an inert argon atmosphere. The spectroelectrochemical measurements of polymer films were carried out in a quartz cuvette, with ITO glass as the working electrode (electrode area 1 cm^2^), a platinum wire as the counter electrode, and a silver chloride-coated silver wire as the pseudo-reference electrode. The potential of the pseudo-reference electrode was −0.3 V versus the Ag/AgNO_3_ electrode. The measurements were performed in 0.1 mol dm^−3^ Et_4_NBF_4_ acetonitrile solutions. UV–VIS spectra of polymer films were registered in situ at various applied potentials corresponding to the films in different oxidation states. Spectra were recorded in the range of 300–1000 nm at fixed potentials during stepwise oxidation from −1.0 V to 0.8 V. All spectra were recorded after reaching the equilibrium state (electrode was conditioned for 300 s at the given potential).

### 2.7. Operando Conductance

Operando conductance measurements were performed using a bipotentiostat setup. For these measurements, interdigitated array platinum electrodes were used, which consist of two interdigitated current collectors, allowing the conductance of the sample to be studied, which is coated on the electrode surface, thus creating electrical contact between the digits. By cycling the potential using a standard cyclic voltammetry procedure while applying a fixed potential bias to the current collectors comprising the interdigitated electrode, information on the conductance of the sample can be obtained during real-time oxidation and reduction processes.

The fixed potential bias (potential difference between the working electrodes) was *E*_bias_ = 10 mV and 20 mV. Operando conductance measurements of PEDOT:PSS and PEDOT:PSS/Salt films were performed in 0.1 mol dm^−3^ Et_4_NBF_4_ acetonitrile solutions. The potential was cycled at 10 mV s^−1^ between −1.5 and 0.8 V. The conductance *G* (S) was calculated using Equation (1) [31].
(1)$$G=\frac{∆I}{2E_{\mathrm{bias}}}$$
where ∆*I* (A) is the difference between the currents on working electrodes, and *E*_bias_ (V) is the bias potential.

All studies in this work were performed at room temperature (20 ± 2 °C).

## 3. Results

### 3.1. Cyclic Voltammetry

Figure 2 shows the CV for the control sample without additives (designated as H_2_O), and for films with various salts additives. Table 1 presents the main characteristics obtained by analyzing CV plots of the films: open circuit potential (OCP, V), cathodic (*E*_c_) and anodic (*E*_a_) peak potentials (V), and specific capacitance *C* (F g^−1^).

The presented data show that the use of different salt additives leads to a noticeable change in the shape of the CV. For films with NaAc or KCl additives, a distorted parallelogram without distinct peaks is observed. In contrast, for a number of other salts, the classic [32,33] shape for PEDOT films—with broad peaks transitioning to a plateau at positive potential values—is observed. Additionally, changes in the basic characteristics extracted from the CV can be seen. The variation in OCP can be attributed to the changes in the doping level of the films in their initial state after coating. The shift of peak potentials and the difference in potentials between the peaks indicate changes in the energy of the redox transitions in the films and changes in the kinetics of charge transfer and mass transfer [34]. Finally, significant changes are observed in the specific capacitance values. The use of NaClO_4_ or LiCl allows for an almost five-fold increase in capacitance compared to films with NaAc or KCl additives. The obtained capacitance values for samples with NaClO_4_, LiCl, and MgCl_2_ additives are close to previous results for PEDOT:PSS films with LiClO_4_ additive in PC solution [27].

It should also be noted that a reference sample without additives displayed a unique dynamic (Appendix A), with capacities increasing during the first cycles. This could be caused by the slow swelling of PEDOT:PSS/H_2_O films in acetonitrile. Such behavior was not observed for PEDOT:PSS/Salt samples, indicating a notable change in electrochemical properties even for samples with similar capacitance values.

For a better understanding of the variation in kinetic characteristics, measurements were performed at different scan rates. The effect of the scan rate on the CV is shown in Figure 3a,b for some representative samples. It can be seen that for the sample with the addition of NaAc (Figure 3b), there is a smaller increase in currents and a noticeable increase in the slope of the CV. Figure 3c shows the dependence lg(*I*_p_)–lg(*v*) for PEDOT:PSS/NaAc and PEDOT:PSS/LiCl. The slope of this dependence enables the estimation of the limiting stage of the process according to Equation (2).
(2)$$I=av^{b}$$
where *I* is the current (A), *v* is the scan rate (V s^−1^), *a* and *b* are numerical coefficients.

The coefficient *b* may take values from 0.5, which corresponds to the diffusion-controlled process, to 1, which corresponds to the surface redox reaction [35]. Table 2 shows the values of the coefficient *b* for the investigated films for both anodic (*b*_a_) and cathodic (*b*_c_) processes. For most of the samples, the values of the coefficient *b* correspond to mixed kinetics, which suggests insignificant limitations of mass transfer in the films. At the same time, the displacement of the peak potentials as the sweep rate increases (Figure 3a) suggests the presence of kinetic limitations of the electron transfer rate. A noticeable contribution of limited mass transfer was observed for only a few samples, namely, PEDOT:PSS/NaAc and PEDOT:PSS/KCl, with values of *b* close to 0.5.

### 3.2. Morphology and Composition

What may account for such significant variations in electrochemical activity? One common hypothesis is a change in surface morphology, an increase in roughness due to phase separation in the dispersion, and changes in deposition conditions [36,37]. The morphology of the films was studied by SEM, and the images of several characteristic films are shown in Figure 4. PEDOT:PSS (H_2_O) (Figure 4a) films have a dense, homogeneous surface morphology.

The addition of salts contributes to a more irregular and rough structure. However, by varying the type of salt, a wide range of different micrometer-scale rough structures can be observed. For some samples, such as PEDOT:PSS/LiCl (Figure 4d), fairly smooth films with “veins” of a few micrometers in size are observed. For other films, such as PEDOT:PSS/NaAc (Figure 4c) or PEDOT:PSS/MgCl_2_ (Figure 4b), a highly rough surface with numerous inhomogeneities is observed. It should be noted that for films with salt additives, an additional change in morphology may arise due to salt crystallization during water evaporation (especially noticeable for the PEDOT:PSS/NaCl and PEDOT:PSS/KCl samples (Appendix A)). Analysis of the data obtained from CV and SEM does not indicate clear correlations, for example, between surface non-uniformity and capacitance. Thus, addition of NaCl or KCl caused a marked change in morphology, making the films very irregular; however, their capacitance was low, while addition of LiCl led to fairly regular films and high electrochemical activity. At the same time, the addition of MgCl_2_ also resulted in high capacitance, but contributed to the non-uniformity of the films.

Another possible reason for the variation in the electrochemical activity may be a change in the surface composition of the films. In many studies, it has been shown that treating the films reduces the concentration of PSS on the surface, which can lead to an increase in conductivity [38,39]. The surface composition of PEDOT:PSS films was studied by XPS. Figure 5 displays high-resolution XPS spectra for several samples in the S2p region. For all samples, typical spin-split doublets of S 2p_1/2_ and S 2p_3/2_ at about 168 eV corresponding to SO_3_− groups of PSS and about 165 eV corresponding to thiophene sulfur atoms in PEDOT are detected.

The ratio between the peak areas allows estimating the ratio between the PEDOT and PSS content on the surface of the films. The concentration of PSS dopants in PEDOT is estimated to be 2.1:1 for PEDOT:PSS/H_2_O, 3.5:1 for PEDOT:PSS/KCl and PEDOT:PSS/NaNO_3_, 3.0:1 for PEDOT:PSS/NaAc, 2.3:1 for PEDOT:PSS/LiCl, and 6.5:1 for PEDOT:PSS/NaClO_4_. For all samples, the concentration of PSS on the surface increases, compared to the pristine films, but these changes do not correlate with changes in capacitance values. Thus, the capacitance values of PEDOT:PSS/LiCl and PEDOT:PSS/NaClO_4_ are similar, although the PSS concentration varies significantly.

For samples with salt additives, a shift in SO_3_-group peaks was also observed. This may correspond to an increase in the concentration of deprotonated SO_3_-groups bound to metal cations [40]. The introduction of cations and anions of salts into the film is detected on the surface by the appearance of new signals from nitrogen, chlorine, or potassium for the corresponding samples (Appendix A).

### 3.3. Electrochemical Impedance Spectroscopy

To better understand the difference in electrochemical properties of PEDOT:PSS and characterize charge transport, the films were investigated using EIS at different potentials. Figure 6 displays Nyquist plots for PEDOT:PS/NaClO_4_ and PEDOT:PSS/NaAc samples. Most of the samples exhibited behavior similar to PEDOT:PSS/NaClO_4_ (Figure 6a).

At a potential of −1.0 V, a semicircle with high resistance was observed, indicating low film conductivity in the reduced state. As the potential increases, the shape of the spectra changes dramatically. The impedance values decrease by more than an order of magnitude. For some samples, such as PEDOT:PS/NaClO_4_, a characteristic “hockey-stick” form is observed, which is typical for the impedance of conducting polymers and porous systems [41,42,43]. A short linear Warburg response is observed in the middle frequency range, and then a capacitance-type response with an angle close to 90° is observed at low frequencies, attributed to the capacitance of the film (*C*_lf_). Simultaneously, for some samples in the middle frequency range, a semicircle is observed, likely corresponding to the charge transfer process at the film–solution interface. The diameter of this semicircle is equal to the charge transfer resistance (*R*_ct_), and for some films at 0.0 V, it was 275 Ω (LiCl), 510 Ω (H_2_O), 270 Ω (NaCl), 150 Ω (MgCl_2_), 2170 Ω (KCl), or 4335 Ω (NaAc). The obtained *R*_ct_ values correlate well with the CV data at different scan rates.

Among all the investigated samples, PEDOT:PSS/NaAc (Figure 6b) is remarkable, for which at −1.0 V, a semicircle is observed in the medium-frequency region, and in the low-frequency region, a linear response is observed. When the potential is increased, the total impedance value decreases by only 2–3 times. This may indicate an incomplete transition to the oxidized form, consistent with the low redox activity of PEDOT:PSS/NaAc and the absence of pronounced peaks in the CV. Similar dynamics were also demonstrated by the PEDOT:PSS/KCl sample.

To study the electron and ion transport processes in the films, binary diffusion coefficients were calculated based on the Mathias–Haas model [44] according to Equation (3).
(3)$$D=\frac{1}{2}{\left(\frac{\left(t_{e}^{2}+t_{\mathrm{ion}}^{2}\right)h}{2\sigma_{w}C_{\mathrm{lf}}}\right)}^{2}$$
where *D* is the binary diffusion coefficient (cm^2^ s^−1^), *t*_e_ and *t*_ion_ are the electron and ion transfer numbers, respectively (assumed here that *t*_e_ = *t*_ion_ = 0.5), *h* is the film thickness (on average about 1 µm based on micrometer measurements, SEM, and mass and density estimates), σ_w_ is the Warburg constant (Ω s^−0.5^), and *C*_lf_ is the low-frequency capacitance (F). The value of *C*_lf_ was estimated from the slope of the dependence −*Z*_Im_–ω ^−1^ (Appendix A). The obtained capacitance values were lower than those obtained by cyclic voltammetry, which is typical when comparing the results of these methods for conducting polymers (Appendix A). At the same time, the relationship between the low-frequency capacitance and the nature of the additive appeared to be similar for the two methods (for example, the lowest capacitances were obtained for PEDOT:PSS/NaAc and PEDOT:PSS/KCl, and the highest for PEDOT:PSS/LiCl and PEDOT:PSS/NaClO_4_, indicating good agreement of the measurements. The values of the Warburg constants were obtained from the slope of the parallel linear *Z*_Re_, *Z*_Im_–ω ^−0.5^ dependences (Appendix A).

The obtained logarithmic values of the binary diffusion coefficients are shown in Figure 6c,d. Analyzing the results for the calculated values of the binary diffusion coefficient, it can be concluded that most additives led to a decrease in the ionic conductivity of the films. This is not an unambiguous conclusion, since the separation of ionic and electronic conductivities for this type of materials when changing the doping level is a complex problem, but earlier results indicate that the electronic conductivity of such materials is 3–8 orders of magnitude higher than the ionic conductivity [45,46]. Therefore, in the first approximation, we can say that changes in binary diffusion coefficients will occur primarily due to changes in the ionic component. A similar effect of decreasing ionic conductivity when DMSO is added to PEDOT:PSS was previously shown [47]. At the same time, DMSO is one of the most well-known additives that increases the electronic conductivity of PEDOT:PSS. The authors interpret this as a change in film structure that promotes electronic conductivity but at the same time impedes ion transport in the film.

To gain a deeper insight into the difference in charge transport in the film, we conducted operando conductance measurements of polymer films in solutions combined with cyclic voltammetry.

### 3.4. Operando Conductance

The measurements were made at *E*_bias_ equal to 10 mV or 20 mV, and the values of ∆*I* changed linearly according to Ohm’s law. The obtained dependences of conductance on potential are shown in Figure 7.

According to conductance changes during cycling, samples can be divided into several groups. In the first group typical for PEDOT [26,48], a plateau of conductance at the anode potentials, and conductance falling practically to zero (about 10^−6^ S) at the transition to negative potentials, were observed (NaClO_4_, NaNO_3_, LiCl, H_2_O). The typical hysteresis for forward and reverse scans was also observed [48]. The lowest hysteresis (on the order of 450 mV) was observed for the PEDOT:PSS/NaNO_3_. Changes in the transition potentials from conducting to non-conducting states and vice versa, as well as changes in the hysteresis values with variations in the nature of the salts, correlate well with changes in the peak potentials and the difference in potentials between the peaks in the CV (Figure 2 and Table 1). PEDOT:PSS(NaAc) differs from other samples in that its conductance remains almost constant over the whole range of potentials studied (maximum decrease of 3%). This behavior is unique to PEDOT:PSS films and may indicate a “freezing” of the redox activity. The other samples occupy an intermediate position, where a decrease in conductance is observed in the region of negative potentials, but not to zero values. The decrease in conductance relative to the values at positive potentials was 10% for PEDOT:PSS/KCl, 22% for PEDOT:PSS/NaCl, and 45% for PEDOT:PSS/MgCl_2_. The degree of decrease in conductivity of the films during their reduction can be used to estimate the completeness of the redox transformation. EIS data at different potentials agree well with the patterns described above, such that for PEDOT:PSS/NaAc and PEDOT:PSS/KCl the difference in impedance was smallest at −1.0 V and 0.5 V, which also indicates incomplete reduction of the films.

In addition, it should be noted that for most of the samples the maximum conductance at the anode potentials remained constant during cycling, but for the PEDOT:PSS/LiCl and PEDOT:PSS/H_2_O samples a decrease in conductance from cycle to cycle was observed. At the same time, the electrochemical activity changed insignificantly.

### 3.5. In situ UV-VIS Spectroelectrochemistry

The transition from a conducting state to a non-conducting state can also be detected by changes in the absorption spectra of the films, since PEDOT is characterized by electrochromism in the visible region of the spectrum, changing from a dark-blue non-conducting state to an almost transparent conducting state [49]. A typical view of the UV-visible absorption spectra at different potentials of PEDOT:PSS electrodes is shown in Figure 8a (sample PEDOT:PSS/H_2_O).

In the reduced state (−1.0 V) there is a pronounced peak with a maximum around 630 nm corresponding to π → π* transitions of the neutral form of PEDOT:PSS. The presence of pronounced shoulders near λ_max_, as well as a low-intensity shoulder extending into the UV region, is commonly attributed to the presence of various crystalline and amorphous structures in the films [50]. It is more convenient to evaluate the dynamics of the spectra with changes in potential in the form of differential spectra relative to the spectrum at −1.0 V (Figure 8b). As the potential increases, the intensity of *λ*_max_ = 630 nm decreases, and at the same time there is an increase in intensity in the region above 750 nm, with the formation of a new intermediate absorption band with a maximum around 880 nm. As the potential is further increased, the intensity of this band decreases and the band with a maximum in the IR region grows. These bands are traditionally attributed to the absorption of oxidized PEDOT fragments (polarons and bipolarons) [51,52,53]. The differential spectra also easily capture the typical PEDOT set of isosbestic points around 770 nm and 850 nm.

For most of the samples, the dynamics of changes in the spectra are similar, and the key difference is observed in the change in intensity of the band with *λ*_max_ 630 nm. The ratio of the intensity at 630 nm at a potential of −1.0 V to the intensity at 0.8 V was 4.1 for H_2_O, 3.8 for NaClO_4_, 4.0 for LiCl, 3.5 for NaNO_3_, 2.6 for MgCl_2_, 1.8 for NaCl, 1.6 for KCl, and 1.3 for NaAc. The sample PEDOT:PSS/NaAc (Figure 8d) is characterized by the weakest change in band intensity at 630 nm, i.e., the film remains predominantly in the oxidized state even at −1.0 V. A small change in the intensity of the 630 nm band was also observed for samples with NaCl, KCl, and MgCl_2_ additives. Unfortunately, it is difficult to estimate the change in band intensity corresponding to polaron and bipolaron bands (maximum around 1600 nm, based on literature data [54,55] due to the limited measurement range). However, even indirect results for the change in band intensity of neutral fragments agree with the operando conductance measurements. The data obtained are consistent, with less variation in absorption band strength seen for samples containing NaCl, NaAc, KCl, and MgCl_2_, and for each of these samples no sharp decrease in conductivity was observed during film reduction. Comparing the other samples is more challenging because the absorption spectroscopy method does not provide as much new detail when potentials are changed. At the same time, the small differences in the spectroelectrochemical data between PEDOT:PSS/H_2_O and some samples such as NaClO_4_ or LiCl are in good agreement with recent work [50].

## 4. Discussion

From the analysis of the data we obtained, the nature of the salt has a noticeable effect on the electrochemical properties of PEDOT:PSS films. However, it is better to choose a more characteristic quantitative parameter for a more detailed analysis of the observed dependencies. Capacitance is the best choice, since it varies significantly for all samples and has practical importance.

To test the hypotheses formulated, the list of salt additives used was extended. Appendix A shows the list of salts and CV parameters. First, it was found that simple patterns such as cation/anion charge or size changes could not explain the observed patterns of capacity changes. Several regularities are currently known for the effect of salt addition or post-treatment with salts on the conductivity of PEDOT:PSS films, as well as some hypotheses used by the authors to interpret the results. For cations, one of the main hypotheses is that conductivity depends on softness parameters [20,21]. Higher conductivity values were observed for softer cations (Cu^2+^ and In^3+^). Using the softness parameters for the capacitance dependence, we observe a relatively weak inverse correlation between these properties (the correlation coefficient is −0.23) (Figure 9a).

For anions, the softness parameters were not representative of the change in conductivity, nor are they suitable for capacitances in this work (the correlation coefficient is 0.5). (Figure 9b). An alternative hypothesis is the dependence of the conductivity on the p*K*_a_ values of the acids corresponding to the anions [20], which seems more like a random correlation with no obvious physical meaning. This may also be supported by data from [22], where the correlation between conductivity and p*K*_a_ was not observed for lithium salts. The correlation coefficient obtained for the *C*–p*K*_a_ dependence was −0.68 (Figure 9c). This value can be evaluated as a weak relationship, but the large scatter of individual values from the linear dependence is more likely to indicate chance.

Since the capacitance of PEDOT:PSS films clearly depends on the nature of the salt, this effect should be attributed to specific ionic effects. The best-known empirical pattern for specific ion effects is the Hofmeister series (Figure 10) [28], which was first demonstrated for protein precipitation and has been applied to many physicochemical property changes, from the hydration enthalpy of ions in aqueous solutions to the S_N_2 reaction rate or the activity of human rhinovirus [56].

Furthermore, for polyelectrolyte complexes (PEC) such as poly(diallyldimethylammonium)polystyrene sulfonate (PDADMA:PSS), low molecular weight anion introduction was shown to follow the Hofmeister series [57]. Since PEDOT:PSS can also be considered as a polyelectrolyte complex, similar dependencies can be expected. Indeed, if we analyze the capacitance values for films with different salts, the resulting series agrees well with the Hofmeister series. Despite the fact that the Hofmeister series was discovered at the end of the 19th century, there is still no theory that unambiguously explains the observed phenomena. There are many different theories [29,58,59], but it is not the purpose of this paper to consider them, so we limit ourselves to an attempt to verify the correlation of the capacitance with the Hofmeister series. To verify our hypothesis, we used the descriptor proposed in [56], i.e., the radial charge density of the ion (ϸ), determined via Equation (4).
(4)$$ϸ=\frac{q_{\mathrm{ion}}}{\sqrt[{3}]{r_{\mathrm{ion}}^{3}}}$$
where *q*_ion_ is the partial charge of the atom and $\sqrt[{3}]{r_{\mathrm{ion}}^{3}}$ is the effective radius of the ion. Using the values of ϸ for the ions analyzed, correlation coefficients of 0.78 for anions and 0.96 for cations were obtained. Thus, the Hofmeister series hypothesis showed the best agreement with the obtained results. According to the Hofmeister series, the highest capacitance values were obtained for lithium perchlorate and lithium tetrafluoroborate (Appendix A). To the best of our knowledge, the use of the Hofmeister series to interpret the properties of PEDOT:PSS films has not been published before.

These effects may be related to changes in the equilibrium of the polyelectrolyte complex and salts in the dispersion [60]:M^+^_s_ + A^−^_s_ + Pol^+^Pol^−^_PEC_ ↔ Pol^+^A^−^_PEC_ + Pol^−^M^+^_PEC_(5)
where Pol^−^, Pol^+^, M^+^, and A^−^ are polyanion repeat unit, polycation repeat unit, salt cation, and salt anion, respectively, and the subscripts “s” and “PEC” refer to solution and PEC phase, respectively. Depending on their ratio, which is controlled by this “doping” of salt into the condensed phase from the dilute phase, the components of PECs, polymers, salt, and water regulate an enormous range of physical properties. For various PECs, it has been shown that it is more favorable to introduce ions that are on the right side of the Hofmeister series [57,58]. It has also been shown that for equilibrium (5) the entropic component makes the largest contribution [61].

The introduction of salts leads to a charge screening effect followed by a partial separation of oppositely charged PEDOT and PSS molecules, releasing them from strong coupling and allowing the conformational change of PEDOT molecules in the dispersion. This then leads to a change in the structure and morphology of the film formed by evaporation of water. Unfortunately, this work was not able to establish a clear relationship between the morphology or composition of the films and the electrochemical activity. It is likely that more sensitive measurements will be needed in the future to detect changes in the structure of the films, for example, using GIWAXS or GISAXS methods [62,63,64], as well as molecular dynamics simulations [65,66,67].

The wide range of methods used in this work makes it possible to evaluate the variety of material properties observed when using salts of different types. This allowed us to conclude that the effect of increasing the capacitance cannot be clearly attributed to a change in any one key property. On the one hand, it is a change in the kinetics of the charge transfer process, as evidenced by lower values of *R*_ct_ or hysteresis in the conductance measurements. On the other hand, the change in ion transport processes in the film may have an influence. For PEDOT:PSS/NaNO_3_, small hysteresis in conductivity and low *R*_ct_, but very low values of binary diffusion coefficients, were registered. This probably led to a decrease in the capacitance of such films. Conversely, PEDOT:PSS/MgCl_2_ was characterized by incomplete film reduction in absorption spectra and conductance measurements, but high values of binary diffusion coefficients might have compensated for this and provided quite high capacitance values. This again confirms that PEDOT:PSS, despite its popularity, remains a very complex object with mixed ion-electron conductivity and an accompanying electrochemical reaction.

The approach developed in this work demonstrates the possibility of sufficiently fine-tuning the properties of PEDOT:PSS films by using salts of a different nature. The addition of lithium salts, such as LiClO_4_ or LiBF_4_, is useful when high electrochemical activity is required. If high ionic conductivity is important, which can be useful for electrochemical transistor applications because the transistor transconductance can change substantially with small changes in ion mobility, additives of MgCl_2_ or LiCl halides can be selected. If films with low energy hysteresis at the transition from the conducting to the non-conducting state are required, which can also be of interest in electrochemical transistors, NaNO_3_ additives should be considered. By using additives such as NaAc or KCl, it is possible to almost completely “freeze” the electrochemical activity of PEDOT:PSS and provide high electronic conductivity over a wide range of potentials, which may be of interest when using PEDOT:PSS as a conducting additive in energy storage devices. This may be particularly relevant for anode materials whose operating potential range may be well below the redox transition potential of PEDOT:PSS.

## 5. Conclusions

In conclusion, this study demonstrated the significant effect of different salt additives on the electrochemical properties of PEDOT:PSS films. Through the use of a variety of methods, the study showed that the capacitance, ion transport, and charge transfer processes are affected by the nature of the salt additives. The results indicate that the Hofmeister series provides the best correlation between the properties of the PEDOT:PSS films and the nature of the salts used.

This research also highlighted the complexity of PEDOT:PSS as a material with mixed ion-electron conductivity and the accompanying electrochemical reaction. The developed approach allows fine-tuning of PEDOT:PSS film properties by selecting appropriate salt additives based on the desired application. High electrochemical activity requires the use of lithium salts such as LiClO_4_ or LiBF_4_, while high ionic conductivity may require the addition of halides such as MgCl_2_ or LiCl. NaNO_3_ additive provides low energy hysteresis. NaAc or KCl additives should be considered for sustained electronic conductivity over the entire potential range.

Overall, this work provides valuable information on the influence of different salt additives on PEDOT:PSS films, paving the way for further exploration and optimization of the material for various applications, including electrochemical transistors and energy storage devices.

## Figures and Tables

**Figure 1 polymers-15-02468-f001:**
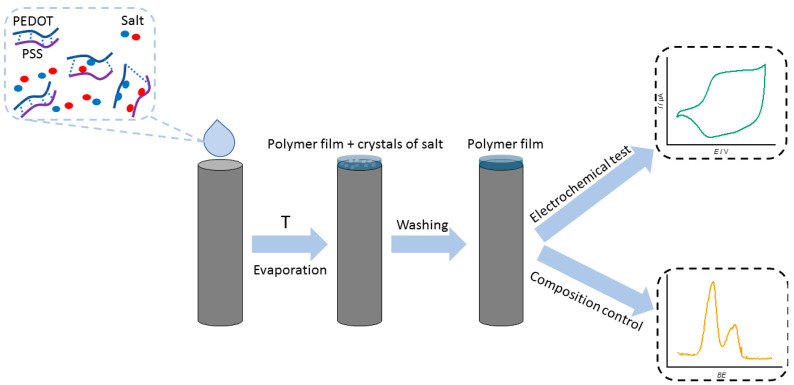
Schematic representation of PEDOT:PSS polymer film fabrication.

**Figure 2 polymers-15-02468-f002:**
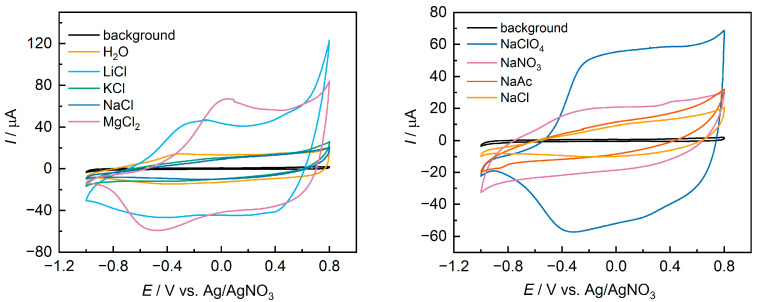
Cyclic voltammograms of studied film electrodes drop-cast from commercial PEDOT:PSS aqueous dispersion with various salt additives. Background curve indicates the response of the pristine GC electrode.

**Figure 3 polymers-15-02468-f003:**
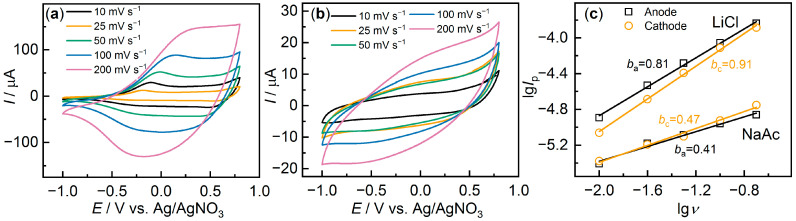
Cyclic voltammetry at different scan rates for (**a**) PEDOT:PSS/LiCl and (**b**) PEDOT:PSS/NaAc; (**c**) lg(*I*_p_)–lg(*v*) dependencies for both samples.

**Figure 4 polymers-15-02468-f004:**
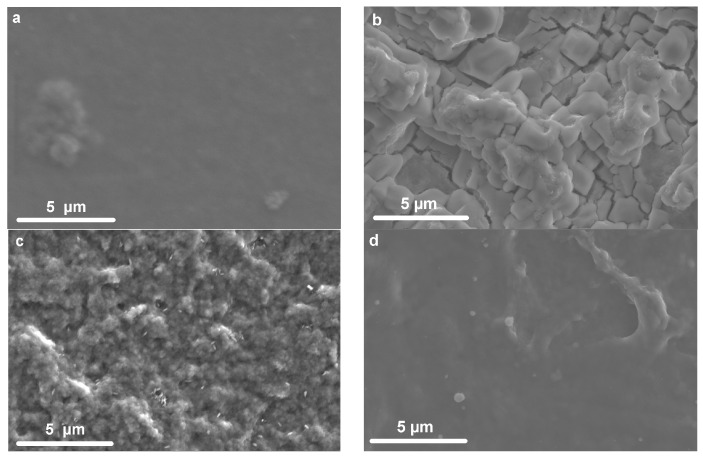
SEM images for (**a**) PEDOT:PSS/H_2_O, (**b**) PEDOT:PSS/MgCl_2_, (**c**) PEDOT:PSS/NaAc, (**d**) PEDOT:PSS/LiCl.

**Figure 5 polymers-15-02468-f005:**
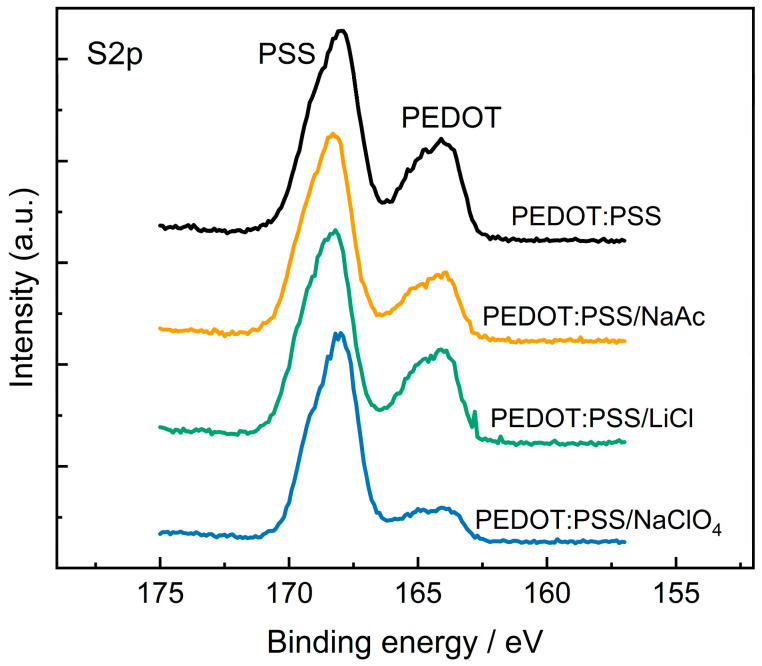
XPS spectra for PEDOT:PSS/H_2_O), PEDOT:PSS/NaAc, PEDOT:PSS/LiCl, PEDOT:PSS/NaClO_4_.

**Figure 6 polymers-15-02468-f006:**
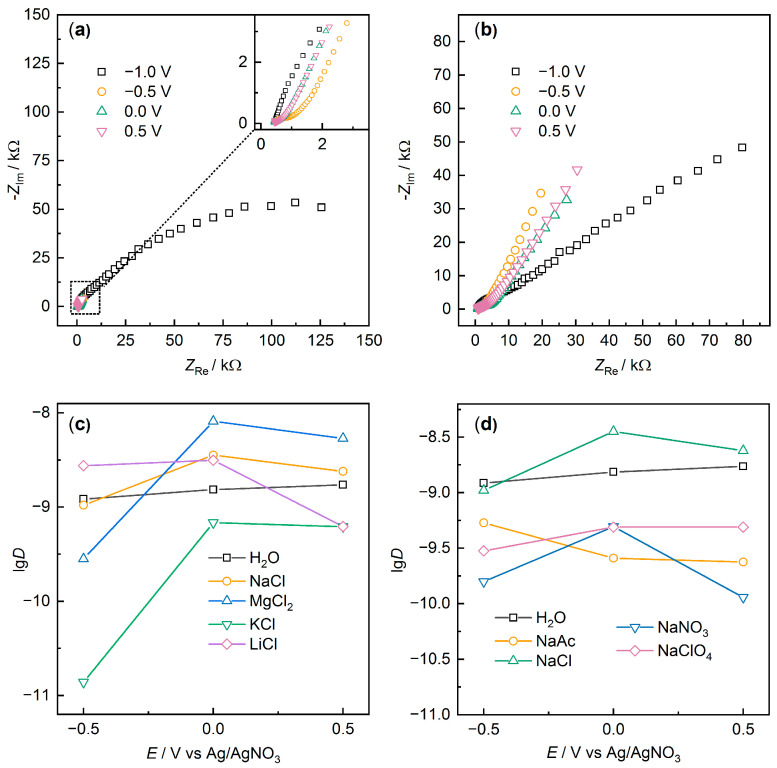
Electrochemical impedance spectra for (**a**) PEDOT:PSS/NaClO_4_, (**b**) PEDOT:PSS/NaAc at different potentials, and (**c**,**d**) lg(*D*) values at different potentials.

**Figure 7 polymers-15-02468-f007:**
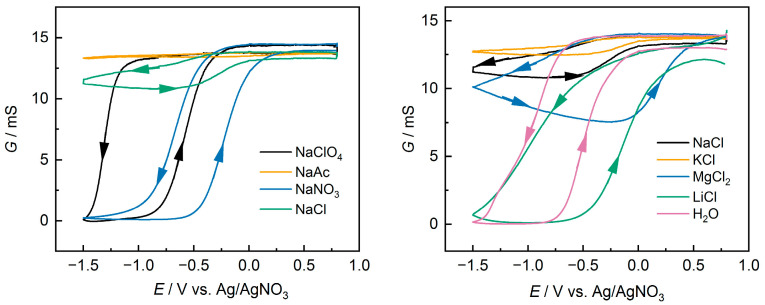
Conductance response as measured by IDE electrode for PEDOT:PSS films at scan rate 10 mV s^−1^; *E*_bias_ = 20 mV.

**Figure 8 polymers-15-02468-f008:**
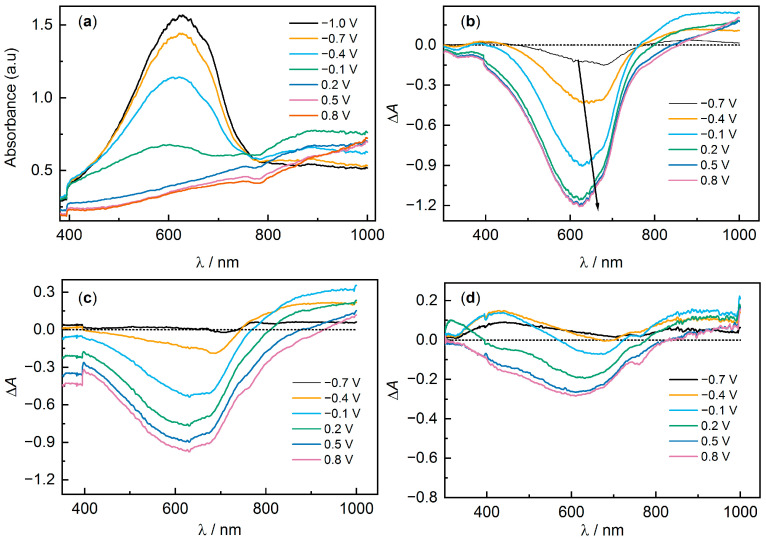
UV-VIS spectra for (**a**) PEDOT:PSS/H_2_O and differential spectra (relative to the reduced form at −1.0 V) for (**b**) PEDOT:PSS/H_2_O, (**c**) PEDOT:PSS/NaNO_3_, (**d**) PEDOT:PSS/NaAc.

**Figure 9 polymers-15-02468-f009:**
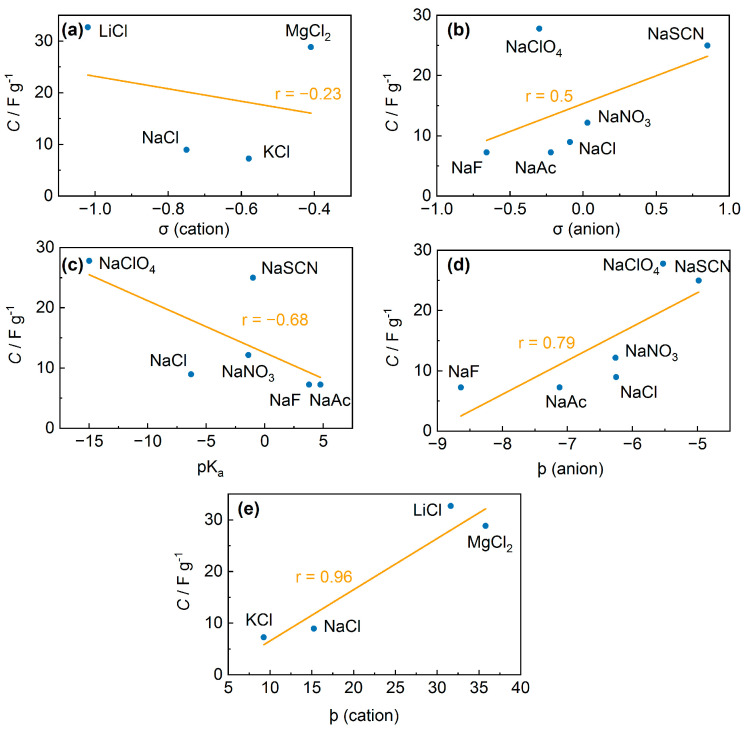
Capacitance of PEDOT:PSS films with additives of salts. (**a**) Capacitance versus softness parameters of anions, (**b**) capacitance versus softness parameters of cations, (**c**) capacitance versus p*K*_a_ values of acids corresponding to the anions, (**d**) capacitance versus radial charge density of the ion (ϸ) for cations, (**e**) capacitance versus radial charge density of the ion (ϸ) for anions.

**Figure 10 polymers-15-02468-f010:**
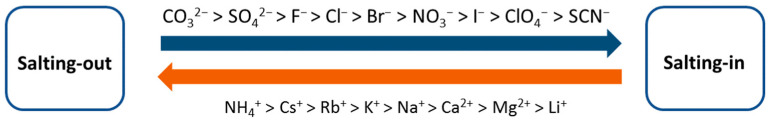
Hofmeister series: ions on the right salt-in (stabilize) the protein, while ions on the left salt-out (destabilize) the protein.

**Table 1 polymers-15-02468-t001:** Electrochemical parameters obtained from CV measurements.

Additive	OCP, V	*E*_a_, V	*E*_c_, V	*C*, F g^−1^
H_2_O	0.1	−0.3	−0.4	8
LiCl	−0.1	−0.2	−0.4	33
NaClO_4_	0.1	−0.2	−0.4	28
NaCl	0.0	−0.1	−0.2	9
NaNO_3_	0.4	—	—	13
NaAc	−0.2	—	—	7
MgCl_2_	0.1	−0.1	−0.6	29
KCl	0.0	—	—	7

**Table 2 polymers-15-02468-t002:** Values of coefficient *b* for anodic and cathodic processes for different salts.

Additive	*ba*	*bc*
H_2_O	0.91	0.87
NaClO_4_	0.74	0.82
NaNO_3_	0.76	0.74
NaAc	0.46	0.47
NaCl	0.71	0.73
KCl	0.63	0.66
LiCl	0.81	0.91
MgCl_2_	0.78	0.74

## Data Availability

The data presented in this study are available on request from the corresponding author. The data are not publicly available due to the policy of the research institution.

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
