# Peer review of "Hofmeister Series for Conducting Polymers: The Road to Better Electrochemical Activity?"

_polymers, 2023, doi:10.3390/polym15112468_

Round 1
Reviewer 1 Report
The proposed work entitled “Hofmeister Series for Conducting Polymers: The Road to Better Electrochemical Activity?” has great scientific value and this can be accepted as soon as possible. Conducting polymers have shown a great potential in energy storage as well as sensor devices. In my opinion this work can add a significant value to the polymer science. I have minor comments that can be considered before publication of this work.
[1] Authors are only discussing PEDOT: PSS polymer. What about other important polymers such as polyaniline, polypyrrole and polythiophene.
[2] Do authors think rest of the polymers can show similar behavior like PEDOT: PSS?
[3] What is the effect of salt additives on porosity of polymer?
[4] Some polymers are electrochemically unstable. Do authors think salt additive can make them stable?
Author Response
Comment 1
Authors are only discussing PEDOT: PSS polymer. What about other important polymers such as polyaniline, polypyrrole and polythiophene.
Response: Thanks for your comments. The polymers like these would also be interesting objects of study. However, the choice of PEDOT:PSS was dictated by the set of its unique properties. First, it could be viewed as a blend of two ionomers: positively charge PEDOT and negatively charged PSS, and it is interesting to observe how the externally incorporated salts would affect the electrostatic interactions between the two. Second, there is a substantial body of work to refer to regarding the conductivity properties of PEDOT:PSS being affected by incorporation of salts and other additives. That means that, on the one hand, the information on electrochemical properties of modified PEDOT:PSS films is lacking, but we could correlate our data with the data on conductivity in the literature and thus expand the knowledge of this material. On the other hand, the data on introduction of additives like these in other conducting polymers you mentioned is lacking even regarding conductivity, and that would require starting a new research project from scratch. In addition, other conducting polymers are usually composed of single chemical species, and thus the laws that govern their properties would be significantly different from those applicable to PEDOT:PSS.
Comment 2
Do authors think rest of the polymers can show similar behavior like PEDOT: PSS?
Response: Thanks for your comments. As we stated above, given that PEDOT:PSS is a combination of two polymers, it is reasonable to expect that the dependencies that exist for it would not be directly transferrable to other polymers. But some similar dependencies can be expected for other polymers with PSS, like PPy:PSS or PANI:PSS. Still, it is an interesting avenue to explore.
Comment 3
What is the effect of salt additives on porosity of polymer?
Response: Thanks for your comments. In this case, we studied the polymers only using SEM, as it is usually sufficient to determine the general morphology of the samples. We found that the pristine polymer is smooth and non-porous, and it sometimes remains that way even with addition of salts, such as LiCl. However, in most cases the introduction of salt additives (such as MgCl2, NaAc or NaNO3) likely provided new “nucleation sites”, so to speak. This allowed the polymer films to dry with the formation of a rough and more porous surface, which might be beneficial for adhesion and film–electrolyte interactions. Regardless, the observed morphology features did not correlate well with the capacitance values, which made it more difficult to link the surface properties to the electrochemical ones.
Comment 4
Some polymers are electrochemically unstable. Do authors think salt additive can make them stable?
Response: Thanks for your comments. That is an interesting question! PEDOT:PSS, which we focus on in our studies, produces stable polymer films, which is excellent for our experiments. This allowed to modify the suspension via multiple ways and to disregard the stability factor, as it should not have been an issue in the first place. However, for other polymers such approach might be an interesting area to explore, especially with the specific goal of stabilization. While using other polymers is out of scope of our research, we assume that changing the electrostatic interactions of the polymer with introduced additives might stabilize the polymer film if the interaction of the polymer with the additive is more preferential to whatever factor would be unfavourable to the polymer’s stability. As we have also shown, the additives affect the electrochemical activity and conductance windows of the films: if this is also the case in other polymers, then by changing the electrochemical activity window, it would be possible to shift the polymer cycling range away from undesirable processes such as overoxidation.

Reviewer 2 Report
I have reviewed the submitted paper entitled “Hofmeister Series for Conducting Polymers: The Road to Better Electrochemical Activity?” submitted to Polymers / MDPI by Apraksin et al. Among a variety of conducting polymers, poly(3,4-ethylene dioxythiophene) doped by poly(styrene sulfonate) (PEDOT: PSS) is well known as the most remarkable conducting polymer in the field. This could be owing to many excellent properties like film-forming, tunable conductivity, thermal and chemical stability, and almost full transparency in the thin film. The materials based on PEDOT: PSS found wide applications including energy storage electrochromic, sensors, etc. Therefore, it is a significant area of research. However, the conductivity for the polymers materials in general is marginal so organic solvents and metal salts have been added to enhance the conductivity. In the submitted paper, a systematic electrochemical study has been made to examine the effect of these additives. This reviewer finds this could be valid and useful to the community in bringing the mechanism involved to divulge the right additive.
The work is well organized (some long sentences can be cut short), and the manuscript is fine with good electrochemical findings/ characterization. The work in its present form is publishable but needs some revisions before rendering a final decision.
The following points need to be considered.
· To mention one typical long sentence, abstract line 19, continues till 22. Like this many occasions are there. Please make it readable and friendly.
· Please define the ZT ratio.
· Line 85 – 87, why the addition of LiClO4 enhances the capacitance, explain in a line or two.
· Line 87, “effect of the type of salt addition” – does this mean cation or anion or both?
· The originality of the work must be a bit more stressed in the last few lines of the introduction. What has not been known until now in the Hofmeister Series for Conducting Polymers?
· Section 2.3, line 130, what is GC? Glassy carbon?
· What is the electrolyte concentration, is at room temp?
· What is operando conductance?
· In Figure 2, background means from the bare electrode without additive?
· Line 203 – 204 shape of CV, charge transfer kinetics can be referred to the similar work reported in the recent literature such as doi.org/10.1021/acsami.0c13755.
· Why the Figure 3b is elliptical?
· Why the SEM images for Fig. 4 a and d are so blurred and masked with a rich film-like domain?
· Has the specific capacitance obtained from Figure 3 CV curves?
· Does the current directly proportional to the CV scan rates relating to the surface-confined redox process,
· Page 15, lines 502 – 503 – why? Please justify the reason why a clear relationship cannot be obtained with the current data.
· The non-electrostatic potentials and ion size, and pH must be correlated.
A few long sentences can be trimmed, otherwise, the overall style is good.
Author Response
Comment 1
To mention one typical long sentence, abstract line 19, continues till 22. Like this many occasions are there. Please make it readable and friendly.
Response: Thank you, we tried our best to make the text more pleasant to read.
Comment 2
Please define the ZT ratio.
Response: Thanks for your comments. ZT is the figure of merit used for thermoelectric systems, describing its efficiency. We are not providing the formula for it that links electrical conductivity, Seebeck coefficient temperature, and thermal conductivity, as it is not studied within the scope of this paper, but we have clarified the definition in the sentence ZT is used. We also clarified the “power factor”, which is also a thermoelectricity-related term.
Comment 3
Line 85 – 87, why the addition of LiClO4 enhances the capacitance, explain in a line or two.
Response: Thanks for your comments. In the study mentioned in this paragraph, the increase in electrochemical activity, i.e., capacitance, was associated with conformational changes in the PEDOT and PSS chains leading to improved charge delocalization when LiClO4 was added. The spectroelectrochemical studies showed that without the addition of the salt, the electron transition energy of the polaron state is higher, which means a lower concentration of charge carriers and thus a lower stored charge. We have updated the manuscript with this explanation.
Comment 4
Line 87, “effect of the type of salt addition” – does this mean cation or anion or both?
Response: Thanks for your comments. It means the effect of the nature of both cation and anion. This point is discussed later in this section.
Comment 5
The originality of the work must be a bit more stressed in the last few lines of the introduction. What has not been known until now in the Hofmeister Series for Conducting Polymers?
Response: Thanks for your comments. Until now, the Hofmeister series application has been largely prevalent in colloidal chemistry and biochemistry. By venturing in this area with PEDOT:PSS, we extend its applicability to electrochemical systems based on conducting polymers. We have rewritten the concluding paragraphs of the introduction to emphasize this idea.
It should be noted, that Hofmeister series are sometimes encountered in the discussion of other conducting polymers (PEDOT or PANI), but when considering them, for example, for use in sensors for the corresponding ions [10.3390/s21010138 ; 10.1016/j.aca.2008.07.001] or as membranes [10.14891/analscisp.17icas.0.i1109.0], or also when studying the mechanisms of electropolymerization [10.1149/1.1432675 ; 10.1039/c9tc00955h]. However, these studies are weakly relevant to our work, both in terms of the object, the type of exposure, and the effect.
Comment 6
Section 2.3, line 130, what is GC? Glassy carbon?
Response: Thanks for your comments. Yes, it is. The abbreviation was defined earlier in section 2.2, yet for clarity of the experimental part, we also added full name here in the revised version.
Comment 7
What is the electrolyte concentration, is at room temp?
Response: Thanks for your comments. Et4NBF4 acetonitrile solutions were always with the concentration of 0.1 M, regardless of the method of electrochemical characterization applied. For preparation of the films, a 1.3% PEDOT:PSS dispersion was diluted with solutions of various salts with the concentration of 0.5 M in 2:3 proportion, resulting in the dispersions with salt concentration of 0.3 M. All measurements were performed at room temperature (20 ± 2 °C). We have updated the materials and methods part.
Comment 8
What is operando conductance?
Response: We appreciate your feedback regarding the lack of detail in Section 2.7. We have since expanded this section in the revised manuscript to better clarify our methodology.
Our procedure is adapted from the method described in the Karlsson et al. article [10.1016/j.electacta.2015.02.193]. It involves the use of interdigitated array platinum electrodes that initially have no electrical contact between their halves. The considerable number of bands (180 pairs) with a minimum gap of 5 μm between them allows the polymer to be applied to the electrode to assess the conductance of the sample, as the sample effectively bridges the digits.
This method is particularly suited for conducting polymers as their conductance varies in response to changes in their doping level. We use a standard cyclic voltammetry procedure and apply a fixed potential bias to the current collectors that make up the interdigitated electrode. This approach allows us to obtain information about the conductance of the sample during real-time oxidation and reduction processes. This real-time feature is referred to as "operando" as opposed to "in situ".
Comment 9
In Figure 2, background means from the bare electrode without additive?
Response: Thanks for your comments. Yes, this refers to the uncoated glassy carbon electrode. We have updated the figure caption to clarify that.
Comment 10
Line 203 – 204 shape of CV, charge transfer kinetics can be referred to the similar work reported in the recent literature such as doi.org/10.1021/acsami.0c13755.
Response: Thank you for your recommendation. We have added a relevant reference.
Comment 11
Why the Figure 3b is elliptical?
Response: Thanks for your comments. The low current values and absence of peaks indicate low redox activity and dominance of double-layer effects, and the noticeable increase in the slope with increasing scan rate indicates noticeable ohmic resistance in the film.
Comment 12
Why the SEM images for Fig. 4 a and d are so blurred and masked with a rich film-like domain?
Response: Thanks for your comments. This is due to the high homogeneity of the film at these magnifications, since more inhomogeneity is observed for other samples, and this effect is weakly visible. For comparison, we present here microphotographs at lower magnifications for these samples (Fig. 1c. in pdf version).
Comment 13
Has the specific capacitance obtained from Figure 3 CV curves?
Response: Thanks for your comments. Specific capacities were calculated from CV at 50 mV s−1, which are represented in Fig. 2.
Comment 14
Does the current directly proportional to the CV scan rates relating to the surface-confined redox process,
Response: Thanks for your comments. Yes, it is a classical interpretation for redox processes for strongly adsorbable Ox and Red forms [Bard, Allen J., Larry R. Faulkner, and Henry S. White. Electrochemical methods: fundamentals and applications. 2nd Edition. Chapter 14. John Wiley & Sons, 2000,, 10.1149/1.1838571]. And this was the case for pristine PEDOT:PSS, yet addition of salts shifted the current responses to the mixed diffusion-controlled process control and surface redox reaction control, to various extent, as suggested by Table 2.
Comment 15
Page 15, lines 502 – 503 – why? Please justify the reason why a clear relationship cannot be obtained with the current data.
Response: Thanks for your comments. In this case, we do not observe a clear correlation between morphology and electrochemical activity. On the one hand, this may be due to the fact that changes in the film structure play a more significant role, for which additional measurements are required, for example, by GIWAXS or GISAXS methods. On the other hand, in situ measurements in contact of the film with the electrolyte are required in this case. Since the SEM was taken for a dry film after application and the electrochemical measurements are performed in dynamic mode for the film in contact with the electrolyte, this causes additional changes in the morphology, which can be decisive in this case. This type of measurement is planned for the future in combination with in situ AFM. We are currently developing a suitable cell for such measurements.
Comment 16
The non-electrostatic potentials and ion size, and pH must be correlated.
Response: Thanks for your comments. Unfortunately, this observation is not very clear to us. But if it is a possible correlation of capacitance with pH or radius of ions, we have considered such hypotheses. Since we mostly use non-hydrolyzable salts, their contribution to the pH change is insignificant. A correlation with ion size is indeed observed, which we use in our hypothesis because the radial charge density of the ion includes the effective radius of the ion as well as its charge.

Round 2
Reviewer 2 Report
I went through the author's responses and the revised part of the manuscript. In this reviewer's opinion, it appears to be reasonably well addressed. Therefore, the revised version is suitable for publication.
Some minor edits are required.